# Robust uncertainty estimates with out-of-distribution pseudo-inputs training

## Abstract

Probabilistic models often use neural networks to control their predictive uncertainty. However, when making *out-of-distribution (OOD)* predictions, the often-uncontrollable extrapolation properties of neural networks yield poor uncertainty predictions. Such models then don't *know what they don't know*, which directly limits their robustness w.r.t unexpected inputs. To counter this, we propose to explicitly train the uncertainty predictor where we are not given data to make it reliable. As one cannot train without data, we provide mechanisms for generating *pseudo-inputs* in informative low-density regions of the input space, and show how to leverage these in a practical Bayesian framework that casts a prior distribution over the model uncertainty. With a holistic evaluation, we demonstrate that this yields robust and interpretable predictions of uncertainty while retaining state-of-the-art performance on diverse tasks such as regression and generative modeling.

## 1 Introduction

Neural networks generally extrapolate arbitrarily [Xu et al., 2020], and high quality predictions are limited to regions of the input space where the networks have been trained. This is to be expected and is only problematic if the associated predictions are not accompanied with a well-calibrated measure of uncertainty. If a neural network is used for estimating such a measure of uncertainty, we, however, quickly run into trouble, as the reported uncertainty then exhibits arbitrary behaviour in regions with no training data. Alarmingly, these are exactly the regions where evaluating the uncertainty is most important to the safe deployment of machine learning models in real world applications [Amodei et al., 2016]. One potential solution is to avoid using directly the output of neural networks for predicting uncertainty, and let it emerge from another mechanism, e.g. an *ensemble* [Hansen and Salamon, 1990, Lakshminarayanan et al., 2017] or some notion of *Monte Carlo* [MacKay, 1992, Gal and Ghahramani, 2016]. Here we explore the alternative view that the networks should simply be trained where there is no data.

But can we train without data? The Bayesian formalism often does so implicitly: most *conjugate priors* can be seen as additional training data [Bishop, 2006], e.g. in Gaussian models, a mean prior $\mathcal{N}(\mu_0, \sigma_0^2)$ can be realised by additional training data of $\mu_0$ with $\sigma_0^2$ setting the amount of observations. Placing a prior over the output of a neural network can, thus, be interpreted as additional training data. Unfortunately, this view is not practical as it implies additional data *for all* possible inputs to a neural network, resulting in infinite data. Our approach is simple: we locate regions of low data density in *input space* and implicitly place observations here in *output space* by min-

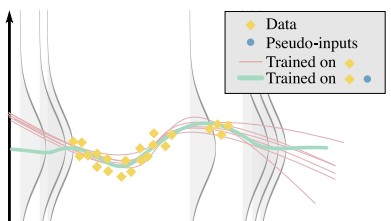

Figure 1: Pseudo-inputs are generated out of distribution, and there we train towards a prior (grey density).

imizing an appropriate KL divergence towards a prior (see Fig. 1). The result is a remarkably simple algorithm that drastically improves uncertainty estimates in both regression and generative modeling.

## 1.1 Background and related work

The predictive performance of machine learning models has drastically increased in the past decade, but the quality of the accompanying uncertainties have not followed. Uncertainties are reported as being miscalibrated [Guo et al., 2017] and overconfident [Lakshminarayanan et al., 2017, Hendrycks and Gimpel, 2016]. Some models even see higher likelihoods of out-of-distribution than in-distribution data [Nalisnick et al., 2019, Nguyen et al., 2015, Louizos and Welling, 2017].

**Neural networks** commonly output distributions which gives a notion of predictive uncertainty. Classifiers trained with *soft-max* is an ever-present example of such. These predictions are generally observed to be *overconfident* [Lakshminarayanan et al., 2017, Hendrycks and Gimpel, 2016] and to carry little meaning outside the support of the training data [Skafte et al., 2019, Lee et al., 2017]. The latter is an artifact of the hard-to-control extrapolation that comes with neural networks [Xu et al., 2021]. In general, since extrapolation is difficult to control, uncertainties predicted by neural networks will exhibit seemingly arbitrary behavior outside the support of the data, yielding untrustworthy results.

**Mean-variance networks** for regression [Nix and Weigend, 1994] model the conditional target density as a normal $p(y|x) = \mathcal{N}\left(y|\mu(x), \sigma(x)^2\right)$ with mean and variance predicted by neural networks. The reported predictive uncertainty is generally accurate in regions near training data, but otherwise unreliable [Hauberg, 2019]. To counter this, Arvanitidis et al. [2017] and Skafte et al. [2019] proposed variance network architectures to enforce a specified extrapolation value, but these heuristics tend to be difficult to tune, and lack principle. Mean-variance networks have seen a recent uptake within generative modeling, where they are applied as an *encoder* distribution in *variational autoencders (VAEs)* [Kingma and Welling, 2013, Rezende et al., 2014].

**Which uncertainty?** A commonly called-upon dichotomy [Der Kiureghian and Ditlevsen, 2009] is that the uncertainty of a model's *prediction* can be decomposed into the uncertainty of the *model (epistemic)* and of the *data (aleatoric)*. The epistemic uncertainty can be lowered by increasing the amount of data, simplifying the model or otherwise reducing the complexity of the learning problem. The aleatoric uncertainty, on the other hand, is a property of the world, and cannot be changed; no prediction should ever be more certain than the uncertainty displayed by the associated data. Depending on the task at hand, we may be interested in different types of uncertainty: In *active learning* [Settles, 2012] and *Bayesian optimization* [Močkus, 1975] we request data for which we have high epistemic, but low aleatoric uncertainty to ensure maximal information gain; while for classification and regression we often just want to minimize the overall predictive uncertainty.

**Bayesian methods** are often used to quantify uncertainty due to their explicit formulation of uncertainty. *Gaussian processes (GPs)* [Rasmussen and Williams, 2005] provide an elegant framework that provide state-of-the-art uncertainty estimates, but, alas, the corresponding mean predictions are often not up to the standards of neural networks. GPs are tightly linked to *Bayesian neural networks (BNNs)* [MacKay, 1992] that place a prior over the network weights and seek the corresponding posterior. Despite advances in *variational approximations* [Graves, 2011, Kingma and Welling, 2013, Blundell et al., 2015], *expectation propagation* [Hernández-Lobato and Adams, 2015, Hasenclever et al., 2017], or *Monte Carlo* methods [Welling and Teh, 2011, Springenberg et al., 2016], training BNNs remains difficult. Furthermore, the predictive uncertainty seems dependent on the degree of approximation and is thus controlled by the available compute power.

**Ensemble methods** have long been used to produce aggregated predictions with uncertainty estimates [Hansen and Salamon, 1990, Breiman, 1996]. *Deep ensembles* [Lakshminarayanan et al., 2017], a collection of differently initialized networks trained on the same data, are generally reported as state-of-the-art for uncertainty quantification in deep models [Thagaard et al., 2020, Ovadia et al., 2019]. As the models in the ensemble are trained on overlapping data, they are correlated, which influence the ensemble uncertainty in ways that remains unclear [Breiman, 2001]. *Monte-Carlo dropout* [Gal and Ghahramani, 2016] casts dropout training [Srivastava et al., 2014] as an ensemble model. It is computationally cheap, but experiments [Ovadia et al., 2019, Skafte et al., 2019] show that the increased correlation of ensemble elements amplifies the method's overconfidence.

**Robustness to distribution shift** is paramount to a well-behaved uncertainty predictor [Ovadia et al., 2019] and must be evaluated accordingly. For out-of-distribution detection, Liang et al.

[2017] proposes a pre-processing perturbation step inspired by adversarial attacks [Goodfellow et al., 2014a] that helps the model distinguish in-distribution and out-of-distribution inputs. Hendrycks et al. [2018] used a *Generative Adversarial Network (GAN)* [Goodfellow et al., 2014b] to generate out-of-distribution pseudo-inputs whose inclusion in the training under an additional regularizing term in the loss function, called *outlier exposure*, enhances the predictor's ability to discriminate out-of-distribution inputs [Lee et al., 2017, Dai et al., 2017].

## 1.2 Robust uncertainty estimates

Our work is strongly inspired by the critical assessment of the issues that undermine variance estimation ran by Skafte et al. [2019] and by the proposal of Stirn and Knowles [2020] which we detail here.

**Notation.** Let the observed variable $x \in \mathcal{X}$ follow the data generating distribution $p_{\text{data}}(x)$, only known through the training dataset of $N$ i.i.d samples $\mathcal{D}_{\text{train}} = \{x_n\}_{n=1}^N$. In the case of supervised learning, the observed variables $x = (x, y)$, with $x \in \mathbb{R}^d$ being the input and $y \in \mathbb{R}^{d'}$ the target for the model, follow the joint decomposition $p_{\text{data}}(x, y) = p_{\text{data}}(y|x)p_{\text{data}}(x)$. The proposed probabilistic model $p_\theta(x)$, whose weights are indicated by $\theta$, aims to accurately emulate $p_{\text{data}}(x)$.

**Practical problems in variance estimation.** Gaussian likelihoods in the form of $p_\theta(x) = \mathcal{N}\left(x|\mu_\theta(x), \sigma_\theta(x)^2\right)$ are widely adopted to model continuous covariates. Real world data cannot be expected to be *homoscedastic*, i.e constant throughout input space, and thus the predictive uncertainty, $\sigma_\theta(x)$, most often uses neural networks to map continuously the observed $x$ onto the parameter space. Beyond the well-known unreliable extrapolation properties of neural networks, this parametrisation of predictive uncertainty is hamstringed by serious defects. Firstly, the predictive variance scales the learning rates of the mean and variance updates by $1/2\sigma_\theta(x)^2$, resulting in a bias for data regions with low uncertainty [Nix and Weigend, 1994]. Secondly, the maximisation of the modeled likelihood is particularly sensitive to scarce data, as local gradient updates for the variance point towards the then undefined *maximum likelihood estimate (MLE)* [Skafte et al., 2019]. Lastly, and more worryingly, such model's likelihood is ill-defined [Mattei and Frellsen, 2018a], as it can arbitrarily and without bound increase when the variance estimates collapse towards a detrimental 0. Overall, the naive maximisation of model likelihood seems insufficient to generate robust and well-behaved uncertainty estimates.

**Student-t likelihood.** The Bayesian formalism, by imposing to learn a parametrised distribution over the predictive uncertainty, offers an attractive view to approaching the problem of uncertainty estimation. Skafte et al. [2019] notably adopts a Gamma distributed precision, $1/\sigma^2 = \lambda \sim \Gamma(\alpha, \beta)$, as the conjugate of an unknown precision for a Gaussian likelihood, to yield a non-standard Student-t distributed marginal likelihood[1]. It is known to offer a more robust likelihood, especially in the scarce data regime [Gelman et al., 2013],

$$p_\theta(x) = \int \mathcal{N}(x|\mu, \lambda)\Gamma(\lambda|\alpha, \beta)d\lambda = T\left(x|\nu = 2\alpha, \hat{\mu} = \mu, \hat{\sigma} = \sqrt{\beta/\alpha}\right). \tag{1}$$

Interestingly, its variance $\text{Var}[x] = \beta/(\alpha - 1) = (\beta/\alpha) \cdot (\alpha/(\alpha - 1))$ can be explicitly decomposed to an aleatoric term $\beta/\alpha$ and an epistemic term[1] $\alpha/(\alpha - 1)$ [Jørgensen, 2020, p16], and offers a direct verification of whether a model knows what it knows.

**Variational variance.** Stirn and Knowles [2020] assumes a latent model precision $\lambda$. This is generated by a prior $p(\lambda)$ and its posterior is approximated variationally by the family of Gamma distributions, conditioned on the inputs to reflect heteroscedasticity. Through *amortized variational inference (AVI)* [Kingma and Welling, 2013] neural networks $f_\phi$ map to the posterior parameters from data, $q(z|f_\phi(x))$. As such, variational variance preserves the modelling capacity and robustness of the non-standard Student-t marginal likelihood, without modifying its parameter architecture, while the definition of a prior over the latent precision induces a more robust training objective. Assuming the likelihood precision is the unique latent code, the *evidence lower bound (ELBO)*,

$$\begin{aligned}\mathcal{L}(q; x) &= \mathbb{E}_{q(\lambda)}\left[\log p(x|\lambda)\right] - D_{\text{KL}}\left(q(\lambda|x) \,||\, p(\lambda)\right) \\ &= \frac{1}{2}\left(\psi(\alpha) - \log\beta - \log(2\pi) - \frac{\alpha}{\beta}(x - \mu)^2\right) - D_{\text{KL}}\left(q(\lambda|x) \,||\, p(\lambda)\right),\end{aligned} \tag{2}$$

takes the form of a regularised log-likelihood, exposing the benefits of the prior regularisation. It penalises predicted variances that would unrealistically get arbitrarily close to either the detrimental

---

[1]See Section I. of the supplementary materials.

limits of 0 or $\infty$, reducing the concerns regarding the ill-definition of the objective. Additionally, the scaling effect of the learning rates of the likelihood parameters is reduced. Naturally, the effect of the regularisation will be highly dependent on the prior selected. Here, because we are mostly interested in enforcing a constant desired uncertainty extrapolation, we adopt an homoscedastic Gamma distributed prior, $p(\lambda) = \Gamma(\lambda | a, b)$, that matches the level of uncertainty observed in data, and leave it for future practitioners to adopt the most adequate prior for the task at hand.

## 2 Out-of-distribution pseudo-inputs training

### 2.1 Dissipative loss

In the variational variance formalism, due to AVI, the predictive uncertainty is controlled by $\alpha$ and $\beta$, the independent neural networks parametrising the posterior distribution, $\mathrm{Var}[\mathrm{x}] = \beta(\mathrm{x})/(\alpha(\mathrm{x}) - 1)$. The unreliable extrapolation properties of neural networks therefore directly challenge the robustness of the method's uncertainty estimates outside of its training support, limiting the applicability of the method. We consider that this flawed extrapolation is not inevitable.

Inspired by outlier exposure [Hendrycks et al., 2018], we propose to include deliberately generated out-of-distribution *pseudo-inputs*, $\{\hat{\mathrm{x}}_k\}_{k=1}^K$ where $\hat{\mathrm{x}}_k \sim p_{\mathrm{out}}(\mathrm{x})$, in the training of our variational objective to constrain the extrapolation of the posterior parametrisation. The optimal variational objective $q^*$ is chosen such that it minimises our proposed *dissipative loss* over the consolidated dataset $\mathcal{D} = \mathcal{D}_{\mathrm{train}} \cup \mathcal{D}_{\mathrm{out}}$, where $\mathcal{D}_{\mathrm{out}} = \{\hat{\mathrm{x}}_k\}_{k=1}^K$,

$$\mathrm{Loss}(q; \mathcal{D}) = -\Big[ \mathcal{L}_{\mathrm{in}}(q; \mathcal{D}_{\mathrm{train}}) + \mathcal{L}_{\mathrm{out}}(q; \mathcal{D}_{\mathrm{out}}) \Big]. \tag{3}$$

The in-distribution component of the loss function $\mathcal{L}_{\mathrm{in}}(q; \mathcal{D})$ naturally arises as the standard ELBO over the training set. The out-of-distribution component $\mathcal{L}_{\mathrm{out}}(q; \mathcal{D})$ operates on a fundamentally different source of data. As the only information available regarding the pseudo-inputs is that they are out-of-distribution, we assert for them a constant, non-informative likelihood $p(\hat{\mathrm{x}} | \lambda) = c$, that has thus no influence on optimisation. This is similar to the strategy of *censoring* [Lee and Wang, 2003] where different likelihoods are used for observations with different properties. As a result, the dissipative loss becomes,

$$\mathrm{Loss}(q; \mathcal{D}) = -\Big[ \sum_{\mathrm{x} \in \mathcal{D}_{\mathrm{train}}} \mathbb{E}_{q(\lambda|\mathrm{x})} \left[ p_\theta(\mathrm{x}|\lambda) \right] - D_{\mathrm{KL}}(q(\lambda|\mathrm{x}) \,||\, p(\lambda)) - \sum_{\hat{\mathrm{x}} \in \mathcal{D}_{\mathrm{out}}} D_{\mathrm{KL}}(q(\lambda|\hat{\mathrm{x}}) \,||\, p(\lambda)) \Big]. \tag{4}$$

It share the same motivating intuition as the *confidence loss* of Lee et al. [2017] and completes the variational variance formalism with a principled mechanism to learn robust variance estimates with the desired extrapolation properties. It indeed explicitly forces the predictor to match our high-entropy prior expectations on out-of-distribution samples while learning the low-entropy covariate dependent distribution, hence the name of dissipative. The reliance of the model's predictive uncertainty on its mean predictions implies that it is primordial here to safeguard its generative performance. We guarantee it with the implementation of a split training procedure [Skafte et al., 2019]; the out-of-distribution regularisation is only applied after the model's mean has been trained.

### 2.2 Pseudo-input generators (PIGs)

Minimising the posterior KL divergence out-of-distribution requires an efficient sampling procedure of pseudo-inputs. As exposed in Fig. 2, their generation should leverage a-priori knowledge about $p_{\mathrm{data}}(\mathrm{x})$ to resolve the undefined nature of $p_{\mathrm{out}}(\mathrm{x})$. In this simple regression case, we show the predictive uncertainty of variational variance models trained on artificial heteroscedastic data. We use a prior uncertainty level that matches the maximum of the data uncertainty. As anticipated, without pseudo-inputs, the model extrapolates uncertainty to a constant, arbitrary level, and only the introduction of pseudo-inputs near the training data results in the desired uncertainty extrapolation. Reassuringly, this suggests that we do not need to regularise our model's extrapolation in the entire out-of-distribution space. Instead, we can focus on the simpler task of generating pseudo-inputs in low-density regions of the input space that neighbours training data, as they can enforce correct extrapolation in the rest of the out-of-distribution space. Lee et al. [2017] gives supporting arguments.

Recent contributions have relied on GANs for generating a useful representation of $p_{\mathrm{out}}(\mathrm{x})$ [Lee et al., 2017, Dai et al., 2017]. Although conceptually intuitive, GANs incur a heavy computational

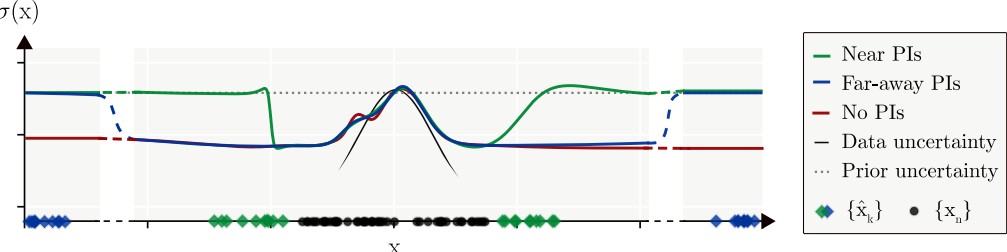

Figure 2: Effect of different pseudo-input distributions on the predictive uncertainty of variational variance models. Training data (black points) is generated uniformly on [-5, 5], with a variance that scales as $\exp\left(-0.5(||\mathrm{x}||/s)^2\right)$. The *near* pseudo-inputs (green diamonds) are generated uniformly in $[-10, -5] \cup [5, 10]$, while the *far-away* (blue diamonds) are on $[-200, -190] \cup [190, 200]$. Dashes amount for the empty space that separates far away pseudo-inputs.

burden and most likely induce serious practical challenges as a result of the instability of their training [Shrivastava et al., 2017]. Furthermore, as one need to understand what is in-distribution to model what it is not, we instead propose to directly leverage the information at hand about the data.

Algorithm 1 gives a simple procedure for generating pseudo-inputs using the data density. Pseudo-inputs are originally sampled from $p_{\mathrm{data}}(\mathrm{x})$, and their positions iteratively updated with gradient descent, with step size $\delta$, by following the directions that minimise their likelihood under $p_{\mathrm{data}}(\mathrm{x})$, similarly to reversed adversarial steps [Goodfellow et al., 2014a].

---
**Algorithm 1:** Pseudo-Input Generator (PIG)
---
$\forall k \in [1, K]$, $\hat{\mathrm{x}}_k \sim p_{\mathrm{data}}(\mathrm{x})$. *iterations* = 0. $\epsilon = \infty$;
**while** *(iterations < max_iterations) & ($\epsilon$ > tolerance)* **do**
    compute $\forall k \in [1, K]$, $\nabla_{\mathrm{x}} p(\mathrm{x})(\hat{\mathrm{x}}_k)$;
    $\epsilon = \max_{k \in [1, K]}(\delta \, \nabla_{\mathrm{x}} p(\mathrm{x})(\hat{\mathrm{x}}_k))$;
    $\forall k \in [1, K]$, $\hat{\mathrm{x}}_k = \hat{\mathrm{x}}_k - \delta \, \nabla_{\mathrm{x}} p(\mathrm{x})(\hat{\mathrm{x}}_k)$;
    *iterations* = *iterations* + 1;
**end**

---

The procedure can run prior to training, in parallel for all $\hat{\mathrm{x}}_k$ with automatic differentiation, and thus results in limited additional complexity for the optimisation[2]. It relies on the availability of a differentiable density estimate of the data, which is, depending on the use case, either directly available (see Sec. 3.2), or can be approximated through a variety of methods such as *Bayesian Gaussian mixture models* [Bishop, 2006], or various *normalising flows* [Rezende and Mohamed, 2015] based methods such as *masked autoregressive flows* [Papamakarios et al., 2017] (see Sec. 3.1). A caveat here is that depending on the PIG's parameters, and on the quality of the density estimate available, pseudo-inputs might be generated in undesired regions of the input space, e.g uninformative density minima. In practice, we adopted conservative density estimates and parameters and did not observe any significant degradation of the predictive uncertainty due to the addition of pseudo-inputs.

## 3   Experiments

**Holistic evaluation of uncertainty estimates**. The ground truth for uncertainty estimates is usually unknown, making their evaluation non-trivial. Similarly as in Stirn and Knowles [2020], we propose to assess them using a collection of metrics. Calibration, which evaluates probabilistic predictions w.r.t the long-run frequencies that actually occur [Dawid, 1982] can be measured by *proper scoring rules* [Lakshminarayanan et al., 2017] such as the model log-likelihood $\log p_\theta(\mathrm{x}|\lambda)$. Additionally, the *root mean squared error (RMSE)* between the predictive and empirical variance, $\mathrm{Var}[\mathrm{x}] - (\mathbb{E}_{q(z|\mathrm{x})}[p_\theta(\mathrm{x}|\lambda)] - \mathrm{x})^2$, offers a quantification of the model's awareness of its own uncertainty. It nevertheless requires an understanding of the model's mean predictive performance, as commonly measured by the RMSE of the mean residuals, $\mathbb{E}_{q(\lambda|\mathrm{x})}[p_\theta(\mathrm{x}|\lambda)] - \mathrm{x}$. We further propose to evaluate the cooperation of mean and uncertainty estimates for the generation of credible samples, which constitutes a consistency check for the learned precision distribution [Gelman et al., 2013], by measuring the RMSE of sample residuals $\mathrm{x}^* - \mathrm{x}$, with $\mathrm{x}^* \sim p_\theta(\mathrm{x})$. Finally, The ELBO, despite the absence of theoretical grounding for it [Blei et al., 2017], is commonly reported as an approximation of the marginal likelihood, and thus of the overall model's predictive performance.

---

[2]Running times are reported in Sec. IV of the supplementary materials.

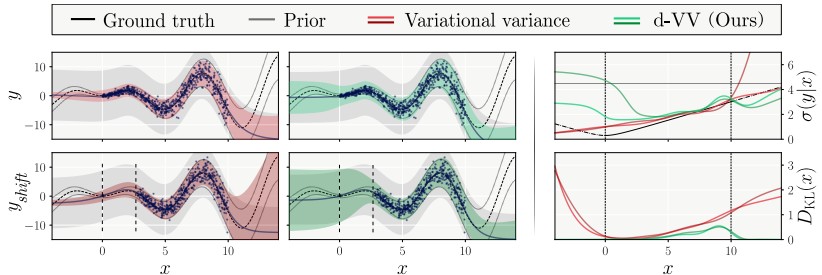

Figure 4: Toy regression results. On the left, the mean predictions are surrounded by $\pm\,2$ standard deviations, with the training data of the bottom row presenting a shift. On the right are displayed the predictive uncertainty fit and the prior KL divergence.

A complete assessment of a model's uncertainty estimates further requires their evaluation under distributional shift [Ovadia et al., 2019], which we either introduce voluntarily through deliberate splitting of the training and test sets, as in Sec. 3.1, or by using test data from a different dataset altogether, as in Sec. 3.2.

## 3.1 Regression

In a regression setting where the proposed model must capture the conditioning between targets and inputs $y|x$, the precision $\lambda$ of a Gaussian likelihood is the only assumed latent code.

Faithfully to variational variance [Stirn and Knowles, 2020] we adopt a Gamma heteroscedastic variational posterior $q_\phi(\lambda|x) = \Gamma\left(\lambda|\alpha_\phi(x), \beta_\phi(x)\right)$ parametrised by the independent $\alpha_\phi$ and $\beta_\phi$ networks, with weights $\phi$, uniquely conditioned on the inputs (see Fig. 3). This approximate posterior, independent of the targets, gives up on the dependency of the true posterior on both covariates to guarantee heteroscedasticity[3].

For strictly more than 2 degrees of freedom, or equivalently, $\alpha_\phi(x) > 1$, the marginal predictive probability $p_{\theta,\phi}(y|x) = T\left(y|2\alpha_\phi(x), \mu_\theta(x), \sqrt{\beta_\phi(x)/\alpha_\phi(x)}\right)$, has its first two moments defined, $\mathbb{E}[y|x] = \mu_\theta(x)$ and $\mathrm{Var}[y|x] = \beta_\phi(x)/(\alpha_\phi(x) - 1)$, providing explicit mean and uncertainty estimates with a single forward

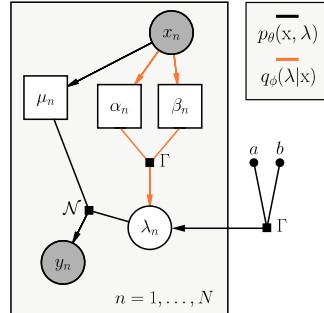

Figure 3: PGM for regression

pass in the single layered, fully connected, $\alpha_\phi$, $\beta_\phi$ and $\mu_\theta$ networks used here. To ensure definition of both the posterior distribution and of the marginal distribution's variance, the parameter maps use a soft-plus activation on their last layer to ensure positivity, and the $\alpha_\phi$ network is further shifted by 1.

The unique dependence of the posterior on the inputs implies that the generation of pseudo-inputs should only rely on the input density. In a general regression setting, it is unknown, and we estimate it here prior to training with a Bayesian Gaussian mixture model [Bishop, 2006]. We refer to it henceforth as *dissipative variational variance (d-VV)*. The specific implementation details are listed in Section II. of the supplementary materials.

### 3.1.1 Toy regression

The desiderata for our method are clear: capture of the data heteroscedasticity, extrapolation to a higher uncertainty level, no underestimation of the predictive uncertainty, and posterior extrapolation to the prior out-of-distribution. Skafte et al. [2019] first showed on the toy regression task, $y = x\sin(x) + 0.3\,\epsilon_1 + 0.3\,x\,\epsilon_2$, where $\epsilon_1, \epsilon_2 \sim \mathcal{N}(0, 1)$, that amongst a collection of methods, only their proposed variance network architecture could realise our first three expectations. Fig. 4 demonstrates that our more principled approach also fulfills all of our requirements, without the need for arbitrarily enforcing the desired extrapolation in our architecture. The importance of out-of-distribution training is also revealed as the standard variational variance approach fails to produce uncertainty estimates that extrapolate correctly and are robust to distributional shift (bottom row of Fig. 4).

---

[3]See Section II. of the supplementary materials for the expression of the true posterior.

Table 1: UCI benchmarks. Each square shows the performance of a given model (rows) on a given dataset (columns). The intensity of the colouring represents the certitude that the associated model performed best on the given dataset. Grey rows mean impossible evaluation for a metric.

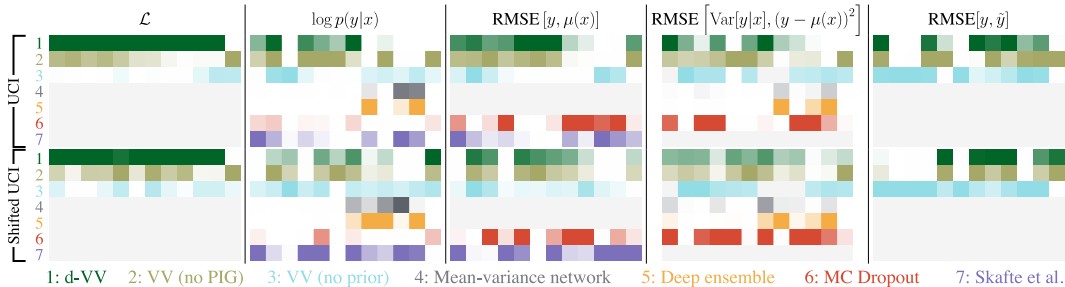

1: d-VV    2: VV (no PIG)    3: VV (no prior)    4: Mean-variance network    5: Deep ensemble    6: MC Dropout    7: Skafte et al.

Table 2: Evaluation of the generative modeling. For each dataset, we report mean ± std over 5 trials.

|  |  | FashionMNIST | SVHN | CIFAR |
|---|---|---|---|---|
| $\log p(\mathrm{x})$ | VAE | $2215.54 \pm 68.81$ | $\mathbf{4304.90 \pm 58.45}$ | $\mathbf{2930.64 \pm 14.82}$ |
|  | d-V3AE | $\mathbf{2349.71 \pm 11.80}$ | $4133.41 \pm 64.28$ | $2668.85 \pm 13.23$ |
| $\mathrm{RMSE}(\mathrm{x}, \tilde{\mathrm{x}})$ | VAE | $0.171 \pm 0.003$ | $0.097 \pm 7\text{e-}4$ | $0.154 \pm 5\text{e-}4$ |
|  | d-V3AE | $\mathbf{0.158 \pm 0.003}$ | $\mathbf{0.087 \pm 0.002}$ | $\mathbf{0.129 \pm 7\text{e-}4}$ |

**Decomposition of the model and data uncertainty.** Fig. 5 presents the decomposition of the predictive uncertainty. The aleatoric component captures the heteroscedastic increase of uncertainty in the training data while the epistemic uncertainty, constant in distribution, extrapolates to higher values. The proposed method therefore demonstrates, to the best of our knowledge, a principled decomposition of uncertainty factors.

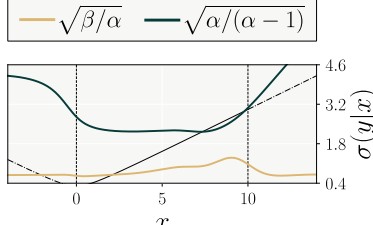

Figure 5: Aleatoric (yellow) and epistemic (dark) uncertainties.

### 3.1.2 UCI Benchmarks

Real world regression datasets from the UCI repository[4] are used to evaluate our model against curated baselines, analogically to the setup from Hernández-Lobato and Adams [2015] and Skafte et al. [2019][5]. As revealed by the summarising Tab. 1[6], our method retains the mean predictive power of the variational variance method. The log-likelihood and RMSE of variance and sample residuals further show the improved calibration resulting from the imposition of a prior on the variance, as the VV methods generally outperform the MLE Student-t (*VV (no prior)*) that shares the same architecture. The table thus proves that holistically, the dissipative loss strengthens the variational variance model's performance, which itself generally surpasses the chosen baselines.

The robustness of the methods to distributional shift is further evaluated as in Foong et al. [2019]. For each input feature, a hole is created in the training data by assigning the middle third of observations to the test set, when sorted w.r.t that feature. Interestingly, we see that our method's calibration slightly improves under the shift, highlighting the robustness benefits of the OOD prior regularisation.

We note that both MC Dropout and the combined method of Skafte et al. [2019] generally perform well, confirming their interest for regression tasks requiring uncertainty quantification, but the former's calibration is not robust to data shifts, as is also reported in Ovadia et al. [2019], and the latter is in practice difficult to tune and lacks principle.

---

[4]https://archive.ics.uci.edu/ml/index.php

[5]See Sec. II. for details about the chosen baselines, datasets and implementations specifics.

[6]The full numbers are included in Sec. II. of the supplements.

## 3.2 Generative models

We extend the evaluation of our proposal to the case of generative models through the lens of VAEs [Kingma and Welling, 2013, Rezende et al., 2014]. Variational auto-encoders infer a low dimensional latent encoding of the data $z \in \mathbb{R}^D$, on which is conditioned the generative process $p_\theta(\mathrm{x}|z)$. Its predictive uncertainty, which evaluates the confidence of the model in its ability to adequately reconstruct inputs is known to be untrustworthy.

In the case of continuous or seemingly continuous inputs, the adoption of a Gaussian decoder $p_\theta(\mathrm{x}|z) = \mathcal{N}\left(\mathrm{x}|\mu_\theta(z), \sigma_\theta(z)^2\right)$ results in an ill-defined model likelihood [Mattei and Frellsen, 2018a] that encourages decoder variance collapse, making the training of the model notoriously harder [Skafte et al., 2019]. Most implementations therefore choose to fix the variance to a set level e.g $\sigma_\theta(z) = 0.1$, or elude the challenge by adopting a Bernoulli likelihood.

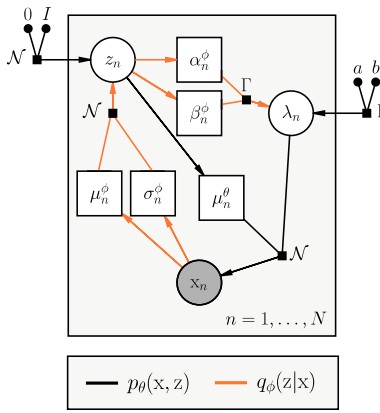

Motivated by our previous results, we now aim to demonstrate that VAEs, whose decoder is fitted with our method, are able to provide robust uncertainty estimates. Assuming a latent generative precision, the latent variables of the model are decomposed into $\mathrm{z} = \{z, \lambda\}$, with $z$ the latent input representations. The marginalisation of the Gamma distributed latent variance results in a Student-T decoder, as detailed in Eq. 1. The overall architecture of the *variational variance variational auto encoder (V3AE)* [Stirn and Knowles, 2020] is shown in Fig. 6, and yields, with the addition of our out-of-distribution pseudo-inputs training, the dissipative loss function[7].

Figure 6: PGM for V3AE

$$\text{Loss}(q_\phi, \theta; \mathcal{D}_{\text{train}}) = -\Big[ \sum_{\mathrm{x} \in \mathcal{D}_{\text{train}}} \mathcal{L}(q_\phi, \theta; \mathrm{x}) - \mathbb{E}_{q_{\text{out}}(z)} \left[ D_{\text{KL}}\left(q_\phi(\lambda|z) \,\|\, p(\lambda)\right) \right] \Big]. \tag{5}$$

Because only the decoder is regularised, the pseudo-inputs lie in the space of latent representations, $\mathcal{D}_{\text{out}} = \{\hat{z}_k\}_{k=1}^K \in \mathbb{R}^D$. The distribution of training inputs is therefore readily accessible as the aggregate variational posterior $q_\phi(z|\mathcal{D}_{\text{train}}) = q_\phi(z|\mathrm{x}_1) \cdots q_\phi(z|\mathrm{x}_N)$. Here again, we rely on a split training procedure to leverage this perk; the encoder parameter maps $\mu_\theta$ and $\sigma_\theta$, as well as the decoder mean $\mu_\phi$ are first trained until convergence, allowing the generation of the out-of-distribution pseudo-inputs and subsequently, the training of the decoder variance.

**Image data.** We evaluate the performance of our proposed *dissipative V3AE (d-V3AE)* against a fully Gaussian VAE on image data, coming from FashionMNIST, SVHN and CIFAR10. For both models, all parameter maps share the same underlying architecture, with the addition of either a softplus and/or a shifting last layer to ensure definition of both the variational and the generative distribution's moments[8].

FashionMNIST

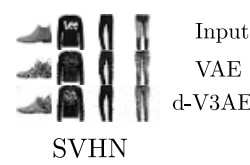

Input
VAE
d-V3AE

SVHN

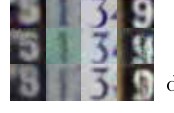

Input
VAE
d-V3AE

CIFAR

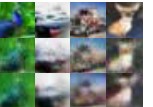

Input
VAE
d-V3AE

Tab. 2 compares model performance on two metrics, the log-likelihood and the RMSE between the original inputs x and reconstructed samples x̃, where $\tilde{\mathrm{x}} \sim p_\theta(\mathrm{x}|\lambda, z)$, $(\lambda, z) \sim q_\phi(\lambda, z|\mathrm{x})$. Unlike most previous implementations, we focus on actual samples, and not the mean, of the generative distributions. This comparison emphasize the cooperation between the decoder's mean and variance, allowing evaluation of the models' uncertainty estimates. Our method both qualitatively (Fig. 7), and quantitatively improves on a Gaussian VAE's sampling ability. The prior smoothens the uncertainty estimates, resulting in more realistic and less crisp samples. The log-likelihoods, evaluated at test time using truncation, i.e. $p_{\text{trunc}}(\mathrm{x}) = p_\theta(\mathrm{x})/(F_{\mathrm{x}}(1) - F_{\mathrm{x}}(0))$, to account for the finite support of data, reveal that our model can achieve a better fit, if the prior is selected correctly. In

Figure 7: Generated samples

SVHN and CIFAR10, the presence of color channels complicates the selection process and challenges our choice of a single homoscedastic prior for all pixels and channels. We note that the dissipative loss also applies to classic VAEs with Bernoulli-only decoders; see Sec. III. of the supplements for details.

---

[7]The derivation of the dissipative loss function is provided in Sec. III. of the supplementary materials.
[8]Implementation details are provided in Sec. III. of the supplementary materials.

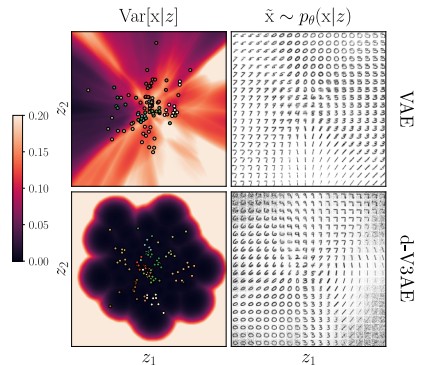

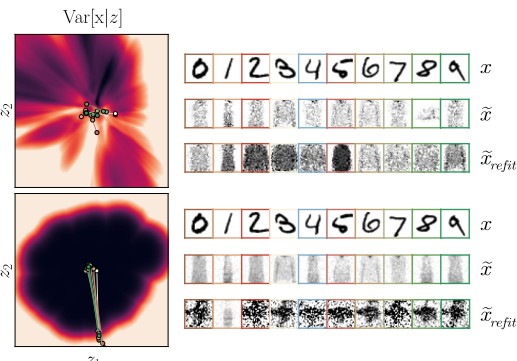

Figure 8: Decoder's aggregated variance (left) and generated samples (right) from the latent space. Coloured points correspond to latent representations of test data, with per-class colours.

Figure 9: Effect of encoder refitting on the latent representations (left) and corresponding samples (right). OOD inputs (first rows, $x$) initially result in in-distribution samples (second rows, $\tilde{x}$). The refitted encoder displaces the encodings (coloured trajectories), modifying the generated samples (third rows, $\tilde{x}_{\text{refit}}$).

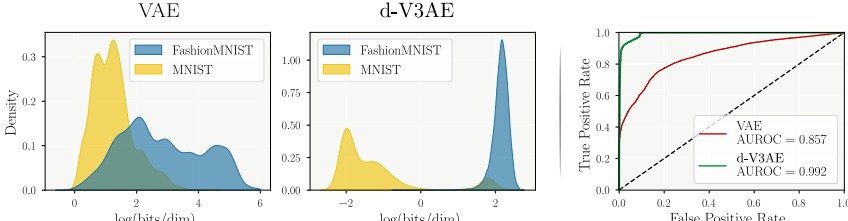

Figure 10: Empirical densities of likelihoods for FashionMNIST (ID) and MNIST (OOD). The clear separation of distributions offered by our method is reflected in the high AUROC shown on the right.

**Applications of robust generative uncertainty.** In Figs. 8 & 9, the colouring of the 2D latent space represent the aggregated decoder variance $\sum_{i=1}^{d}(\sigma_\theta(z)^2)_i$. It is clear that our method displays more regular uncertainty estimates, and provides the extrapolation guarantees we strove for. Beyond increased robustness and better generative power, this unlocks meaningful out-of-distribution detection, beating previous state-of-the-art [Havtorn et al., 2021]. For Figs. 9 & 10, as argued in Mattei and Frellsen [2018b], we refit at test time the encoder of models trained on FashionMNIST on MNIST. The regularity and structure of the decoder variance rewards the encoder for learning to place representations of OOD data outside of the region of in-distribution latent encodings, resulting in a model that is aware of its own inability to reconstruct plausible data, as displayed by the row $\tilde{x}_{\text{refit}}$ of d-V3AE.

## 4   Conclusion

We have introduced a novel loss, the dissipative loss, that leverages artificial out-of-distribution pseudo-inputs for learning robust uncertainty estimates. We demonstrate through a Bayesian approach that casts a prior distribution over the model's variance a principled mechanism for controlling the extrapolation properties of neural networks governing the predictive uncertainty. Our experimental results reflect the benefits of our principled and scalable approach, displaying better calibrated and more robust uncertainty estimates, while matching the predictive power of known baselines. Finally, and most interestingly, our approach can instill into probabilistic models a notion of their own ignorance, increasing their ability to *know what they don't know*.

**Limitations.** The largest limitation of our approach is that it depends on an estimate of the density of the input data. In our experience, even coarse-grained densities are sufficient to significantly improve upon current approaches. However, as one rarely have guaranteed good estimates of the input density, our method cannot be approached as a black-box. One exception seems to be the application to VAEs, where the aggregated posterior, in our experience, always provide a suitable density estimate.

**Negative societal impact.** Improving the ability of predictive models to assess their own confidence is solely a positive contribution as it can help alleviate potential consequences of incorrect predictions. We are therefore not aware of any potential negative impacts of our work.

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
