# OpenReview forum: "Robust uncertainty estimates with out-of-distribution pseudo-inputs training"
_NeurIPS.cc/2021/Conference — NeurIPS 2021 Submitted_

### Official Review · Reviewer_7t1Z · 2021-07-16

**Rating:** 5
**Confidence:** 3

**Summary:**

The paper proposes a method for learnable uncertainty estimation using neural networks without an explicit requirement of known out-of-distribution data. Instead, the model is trained to maximize model fit where there is explicit training data and maximize uncertainty on estimated “pseudo-inputs”. A key contribution of the method is in how the “pseudo-inputs” are generated and crafted. The authors propose a “Pseudo-Input Generator (PIG)” which generates samples based on the training data density -- locations with lower data density will receive more pseudo-inputs. Results are demonstrated on a variety of tasks with increasing complexity ranging from UCI benchmarks to image generation tasks.


**Limitations And Societal Impact:**

Yes

**Main Review:**

**Positives**:

A. Empirical results are both comprehensive (in terms of the datasets and data modalities tested), but also demonstrate strong advances of the proposed method. The authors do a great job at capturing a diverse range of tests, from UCI benchmarks, to synthetic regression data, and higher-dimensional image generation tasks.

B. The result on generative models presented an excellent application of the proposed method beyond only prediction but also towards density estimation. While generative models frequently model aleatoric uncertainty, the ability and demonstration of also epistemic uncertainty is a great contribution.

C. The optimization of placement of pseudo-inputs is a clever approach and contribution. Performing this optimization prior to training is a great observation. Did you experiment with an adaptive approach (updating throughout training) which aims to focus on certain regions of the data space depending on the model’s performance during training.

**Negatives:**

The main two contributions, as stated by the authors, are (1) on how to generate reasonable pseudo-inputs in low-density regions of the input space which neighbors training data and (2) a dissipative loss which can jointly fit the in-domain training data while simultaneously maximizing uncertainty on out-of-distribution extrapolated regions, as determined by the pseudo inputs.

E. I believe contribution (1) is novel and original; however, I think some additional discussion to other works which leverage contrastive priors to emulate the pseudo-inputs would be helpful. An example of such a work is [D. Hafner et al. “Noise contrastive priors for functional uncertainty” UAI 2020]

F. Contribution (2) on the other hand, overlaps significantly with prior work which is not covered in the related work section (despite this section being otherwise very thorough and well written). Specifically I want to raise the author’s attention to the methods of prior networks ([A. Malinin et al. “Predictive Uncertainty Estimation via Prior Networks” NeurIPS 2019]) and evidential learning ([A. Amini et al. “Deep evidential regression” NeurIPS 2020] and [M. Sensoy et al. “Evidential deep learning for …” NeurIPS 2019]). Specifically, the proposed dissipative loss (Eq. 4) is extremely similar to the prior networks training objective which also aims to jointly fit in-domain data while inflating uncertainty on the OOD data. However, a key difference between the proposed method and prior network OOD objective is that prior networks assume the OOD data is given, whereas the proposed method provides an algorithm for generating these. Furthermore, evidential methods (cited above) also leverage in-domain data to “simulate” and act in place of explicit OOD data, but again follow a similar OOD objective to inflate uncertainty on the extrapolated regions. The novelty of the proposed method (and lack of comparisons to both evidential regression and prior networks) in light of these works is my main criticism of the work. Greater discussion explaining the relation to these works would help alleviate my concerns of the contribution.

**Minor**:

G. Figure 3 and 4 are out of order on the page.

H. Add quantitative  x-tick values to Figure 2. The legend references quantitative x-domains but these are meaningless without proper labeling on the plot.


**Time Spent Reviewing:**

4

---

> ### Author Response · Authors · 2021-08-10
> **Reply**
>
> Thank you for your review, and for accurately capturing the benefits of our proposal.
>
> **Did you experiment with an adaptive approach (updating throughout training) which aims to focus on certain regions of the data space depending on the model’s performance during training**
>
> Thank you for the suggestion, we admit we did not experiment with it.
>
> In the case of supervised learning, as the distribution of training inputs is fixed, an adaptive approach would not result in different distributions of generated pseudo-inputs throughout training, but indeed for the generative case, if we were to simultaneously train the encoder and the decoder variance, it would be quite interesting to adaptively re-generate pseudo-inputs throughout training. Our intuition is that it would probably slow down convergence as we could generate pseudo-inputs in not so informative regions of input space, or even in regions that would subsequently become high density regions, resulting in over regularisation.
> Because we chose to adopt a split training procedure, the encoder has fully converged and its weights are fixed when we train the decoder variance, meaning that the distribution of encoded inputs does not in practice change during training of the decoder variance.
>
> **I think some additional discussion to other works which leverage contrastive priors to emulate the pseudo-inputs would be helpful**
>
> Thank you for this suggestion. We admit that we missed this prior work before you and another reviewer provided the reference to us. The method proposed by Hafner et al. is indeed remarkably similar to our proposition, and we refer to our own answer to Reviewer #1 for a detailed explanation of the key differences between their approaches and ours. Overall, we see our proposal as a generalisation of their procedure for generating pseudo-inputs, which we show works best in practice within our variational variance framework.
> Furthermore, because we adopt a censoring approach to the outputs associated with the generated pseudo-inputs, our proposal is not restricted by the strong assumption that they must coincide with the true output associated with the inputs used in the pseudo-input generation.
>
> **Specifically I want to raise the author’s attention to the methods of prior networks ([A. Malinin et al. “Predictive Uncertainty Estimation via Prior Networks” NeurIPS 2019]) and evidential learning ([A. Amini et al. “Deep evidential regression” NeurIPS 2020] and [M. Sensoy et al. “Evidential deep learning for …” NeurIPS 2019]).**
>
> Again, thank you for these references that we had previously missed. The proposed loss of Malinin et al indeed bears resemblance to our dissipative loss as it consists in two terms, the first matching the model with a low entropy distribution on in-distribution data and the second with a high entropy, uninformative prior, but the two terms in their loss adopt the same form, an expected KL divergence over an input distribution. As you pointed out, they must therefore rely on having at hand a complete out-of-distribution dataset over the entire data space ({inputs,outputs}), while we avoid the need to know the outputs associated with the regularising pseudo-inputs through our censoring approach. This distinction is important to understand why it becomes so much easier in our case to generate artificially pseudo-inputs only relying on a density estimate of the inputs, without having to resort to simplifying assumptions wrt the output values attributed to pseudo-inputs as is done for example in the work of Hafner et al.
>
> Furthermore both the models of Amini et al. and Sensoy et al. also regularise their predictive variance with a high uncertainty prior, but do not rely on the addition of out-of-distribution pseudo-inputs to do so. For example, in the case of Amini et al. the prior is enforced where the posterior is misleading, i.e on inputs that result in large errors. So while these methods share a common framework, their objective seems slightly different, as they cannot claim to control the extrapolation properties of their uncertainty predictions.
> Mentioning them in our related works, together with other provided references would nevertheless make sense and enrich our reader’s understanding of the other approaches to regularisation of variance estimates.

---

### Official Review · Reviewer_egD5 · 2021-07-16

**Rating:** 6
**Confidence:** 4

**Summary:**

The authors propose using pseudo-inputs, drawn OOD, during training to address unwanted extrapolation properties.These pseudo-inputs encourage the model to remain close to its prior away from the training data. This algorithm is then evaluated on regression and generative modeling tasks.

**Limitations And Societal Impact:**

Yes, the authors include two sections at the end of the paper to address this.

**Main Review:**

Originality: To the best of my knowledge, the idea of combining neural networks and pseudo-inputs is novel. Section 1, which discusses related work, was especially well written. One suggestion is that the definition of OOD that is used in the paper given Algorithm 1, seems to be input points that have low density under the data distribution. It might be useful to mention this earlier in the paper, when OOD is introduced.

Quality: The regression experiments in 1D (Figs. 1, 2, and 4) are helpful to build intuition, but I did not find the argument in Sec. 2.2 that only some pseudo-inputs near the training data are needed to correctly extrapolate everywhere convincing in higher-dimensional input spaces.

I think an ablation study wrt. to Algorithm 1 or PIGs would improve the paper. There are many ways to generate OOD pseudo-inputs, and it’s not clear whether Algorithm 1 is crucial to the dissipative loss. Similarly the sensitivity to the number of pseudo-inputs is not explored.
Clarity:
- I think the connection between pseudo-inputs and priors in Gaussian processes could be elaborated on to improve the motivation.
- What is the maximum in Algorithm 1 as this is over vectors? How sensitive is the output of the algorithm to the number of iterations and tolerance?
- Table 1 is very difficult to read. It’s not clear which model performed best from the intensity of the coloring

Significance: I like how simple a prescription training on pseudo-inputs is, and the results presented are encouraging. I feel that they would be of interest to the NeurIPS and robust deep learning communities. The significance of the paper would be improved further by benchmarking the method on the OOD calibration tasks of Ovadia et al., but I understand this may be beyond the scope of this work.

**Time Spent Reviewing:**

6

---

> ### Author Response · Authors · 2021-08-10
> **Reply**
>
> Thank you for your appreciation of the conceptual simplicity of training on pseudo-inputs.
>
> **OOD … seems to be input points that have low density under the data distribution**
>
> Yes, that is entirely correct, and we will add this clarification earlier in the paper.
>
> **only some pseudo-inputs near the training data are needed to correctly extrapolate everywhere convincing in higher-dimensional input spaces**
>
> We share your intuition which hints at the difficulty of defining precisely what in practice the notion of “close” and “at the boundary” becomes for pseudo-inputs in higher dimensions. We nonetheless extended the study presented in Figure 2 to higher dimensions. Namely, the following table provides out-of-distribution evaluations of the KL divergence between the variance posterior and prior on two experiments of the UCI benchmarks with respective input dimensions of 4 and 11, for three types of pseudo-input placements. “No OOD” refers to the standard variational variance method (VV) without any pseudo-inputs, “close OOD” refers to our method, with pseudo-inputs generated in regions of low-density of space, and “Far OOD” refers to our method with pseudo-inputs generated as Gaussian noise around training inputs with a large variance (15, for standardised inputs). Very clearly, only our method does not see the out-of-distribution KL divergence increase significantly as the dimensions of the experiment increase, indicating that the posterior extrapolates to the desired prior uniquely for our proposed placement of pseudo-inputs.
>
> | method | uci_ccpp | uci_wine_white |
> |-----------|--------------|----------------------|
> | No ood (vv) | 0.65483451 | 1.71848818 |
> | Close ood (dVV) | 0.01864026 | 0.07170208 |
> | Far ood (gaussian noise, large noise) | 0.02282202 | 0.94072956 |
>
> **the sensitivity to the number of pseudo-inputs is not explored**
>
> In practice for stability reasons (as is most generally done), we implemented our loss using the mean of the respective in-distribution and out-of-distribution components. As a consequence, the method is mostly insensitive to the number of pseudo-inputs used.
>
> **connection between pseudo-inputs and priors in Gaussian processes**
>
> We approached our problem with envy of the nice extrapolation properties seen with GPs, and this served as our “mental baseline” in terms of the properties we want from neural networks. That being said, GP models are technically quite different from the model we investigate. Unlike GPs, we place our prior directly over the output variance. Some work on GPs also consider such constructions (see e.g. Variational Heteroscedastic Gaussian Process Regression, Lázaro-Gredilla & Titsias, ICML 2021) but the resulting models are more complex than usual.
>
> That being said, we do acknowledge that we could improve the paper motivation by calling on intuitions from GPs.
>
> **What is the maximum in Algorithm 1 as this is over vectors? How sensitive is the output of the algorithm to the number of iterations and tolerance?**
>
> We acknowledge that the formulation of line 4 in Algorithm 1 is not clear, the maximum is taken over the norm of the vectors, not on the vectors themselves. The idea is that if the update of all pseudo-inputs through a gradient descent step results in position updates whose magnitude are all below the given threshold, the algorithm should stop, as it implies that all pseudo-inputs are already in low-density regions of the input space.
>
> We generally observed that the algorithm is not very sensitive to the number of steps or the threshold. We refer to our answer to Reviewer #1 regarding our choices of hyperparameters.
> In general, we executed a coarse grid search on a sub-selection of the UCI benchmarks to determine the hyperparameters we then used in all of our experiments. Better results could have been obtained tuning hyperparameters for each individual experiments, as the optimal number of steps and threshold vary from experiment to experiment (see e.g the difference between d-VV and d-VV 0 steps for carbon (0 steps performs best) and naval (5 steps performs best) in the comparison table provided to detail the differences between the work of Hafner et al and ours to Reviewer #1), but because the scale of the difference is relatively small we did not invest the time into it.
>
> **Table 1 is very difficult to read. It’s not clear which model performed best from the intensity of the coloring**
>
> Thank you for this honest feedback, which is seemingly shared by other reviewers. We will provide a table which numerically describes for each method, for each metric, how many of the experiments of the UCI benchmark it performed best at.
>
> **The significance of the paper would be improved further by benchmarking the method on the OOD calibration tasks of Ovadia et al.**
>
> Thank you for this relevant suggestion. The very thorough investigation of Ovadia et al. nevertheless concerns the evaluation of the calibration of the predictive uncertainty of classifiers, which were mostly left out of this study as a consequence of our initial and deliberate decision to focus on Gaussian VAEs, which offer a nice connection to mean variance networks for regression. We nevertheless hope that our work can be extended to classification cases and have, as the annex suggests, started our own investigations with discrete VAEs.

---

> > ### Comment · Reviewer_egD5 · 2021-08-18
> > **Increasing score**
> >
> > I'm grateful to the authors for responding to my questions. The changes discussed here and in response to other reviewers will clearly improve the paper, so I will increase my score. However, it does seem like there were a number of important issues raised that will mean fairly substantial changes to the paper. This includes the paper's relationship to prior work (which I was unaware of) pointed out by Reviewers 23bM and 7t1Z.

---

### Official Review · Reviewer_6v5r · 2021-07-16

**Rating:** 6
**Confidence:** 4

**Summary:**

The authors propose an approach (called d-VV) for learning better uncertainty estimates for neural network models. The core idea is a variation (pun not intended) on the variational variance work by Stirn and Knowles (2020) where an additional "dissipative" term is added to ELBO, which enforces that the latent precision stay close to its prior outside the training distribution as well. This is done to ensure that the uncertainty estimates are accurate even on OOD inputs.

The authors go on to propose a method for emulating "informative" OOD samples to estimate this dissipative term. This is achieved by starting from data samples and taking negative gradient steps on some differentiable density estimate (which is presumed to be available).


**Limitations And Societal Impact:**

The authors addressed these concerns appropriately.

**Main Review:**

The paper's writing is quite good, and it is apparent by the extensive background section that the authors have a good command of the topic at hand. d-VV is well motivated and clearly exposed, and there is a clear effort from the authors to conduct a thorough empirical evaluation on the benchmarks they chose (although the results are not always communicated in the clearest fashion).

To me what holds the paper back is the lack of experiments on more "realistic" benchmarks: the largest datasets are SVHN/CIFAR and the biggest model is a 5 layer CNN. It appears that the main drawback of d-VV is that it necessitates the introduction of auxiliary models to (1) compute the variance through the latent code and (2) yield density estimates for deriving OOD pseudo inputs. Whether the associated computational overhead precludes the use of the method in large scale settings is not apparent in the current version of the paper.

This is why I think the paper is borderline, although I am willing to change my score to a more positive evaluation if some of the cons below are addressed.

### Pros

- Well written, good background and principled approach
- Simple concept
- Good empirical results

### Cons

- Possibly lots of computational
- Only toy-ish experiments, would be interesting to test on more realistic settings with large models
- Visualizations are hit or miss: they rely too much on color. While the holistic evaluation in Table 1 (which is actually a figure and not a table) is interesting, I think this section is missing a "punchline table" with the key results (in numerical form rather than as a heatmap).

### Remarks

- Eq. 4 would be clearer with proper bracketing of the sum over D_train
- L567 in the appendix: "a single hidden layer with 50 hidden layers" (I assume you mean 50 hidden units)




**Time Spent Reviewing:**

4

---

> ### Author Response · Authors · 2021-08-10
> **Reply**
>
> Thank you for your review, and for raising your concerns regarding the scalability of our proposed method.
>
> **Possibly lots of computational**
>
> We recognise that this is a valid concern, density estimation required for the PIG for large scale datasets can be hard. We nevertheless do not find that it holds back the method as the pseudo-inputs can be generated independently,  with a direct control over the computational cost through the number of components of the (Bayesian) Gaussian Mixture used in practice as an approximate density.
> As argued in the paper, this procedure can be executed prior to training as long as the whole distribution of training samples is available. This is frequently the case for supervised learning experiments, and in the case of generative models, through the prism of VAE, a split training procedure is implemented. The encoder mean and variance networks, as well as the decoder mean can be pre-trained as is usually done, unlocking access to the distribution of encodings of training inputs, which can in a following step be leveraged to generate the pseudo-inputs required to eventually train the variance networks of the decoder.
>
> **Only toy-ish experiments, would be interesting to test on more realistic settings with large models**
>
> Influenced by the chosen baselines, we indeed restricted ourselves to experiments whose size does not necessarily represent perfectly realistic settings. We note that only Lakshminarayanan et al. Simple and scalable predictive uncertainty estimation using deep ensembles. NeurIPS, 2017, used larger experiments than CIFAR10/SVHN.
> We argue nevertheless that these experiments already demonstrate the scalability of our approach in terms of dimensions of the space in which pseudo inputs are generated, as we adopted for them a latent space of dimension 126.
>
> We do not see the additional computational burden of our method as prohibitive. Since the training of the variance networks can be split from the training of the mean network our approach scales gracefully. The additional complexity of our method is predominantly that we have to train two networks (alpha and beta), instead of a single variance network. This is not prohibitive for the scalability of the model.
>
> **Visualizations are hit or miss: they rely too much on color. While the holistic evaluation in Table 1 (which is actually a figure and not a table) is interesting, I think this section is missing a "punchline table" with the key results (in numerical form rather than as a heatmap).**
>
> We acknowledge that the heatmap can be improved, and will add a summary table in numerical form that sums up the number of significant “wins” for each method.

---

> > ### Comment · Reviewer_6v5r · 2021-08-31
> > **Response to the rebuttal**
> >
> > I thank the author for taking the time to comment on my concerns.
> >
> > It is true that the computational overhead is somewhat attenuated by the fact that pseudo inputs can be pre-computed. I still think that experiments on more complex datasets or tasks would be helpful: it would be interesting to assess the effectiveness of pseudo inputs in a much richer/diverse datasets or on bigger models where the decision boundary is presumably more complex. In my opinion the fact that previous work has largely stuck to mnist/cifar is not a particularly convincing argument, and since the paper's contribution is largely empirical it stands to reason that the experiments should be more realistic.
> >
> > With that in mind, and considering the concerns raised by other reviewers, I will keep my score the same. I think the paper could be accepted as is, but it would be a much stronger submission by (1) carrying out the changes described in response to other reviewers (especially with regards to comparisons with previous work) and (2) showing empirical results on a more realistic dataset (eg. imagenet).

---

### Official Review · Reviewer_23bM · 2021-07-17

**Rating:** 5
**Confidence:** 4

**Summary:**

This paper considers the idea of generating pseudo-inputs (i.e. extra augmented samples) to improve OOD uncertainty estimates by training with an uncertainty-increasing ("dissipative") loss at these pseudo-inputs. The pseudo-inputs are generated by minimizing estimated density under the data distribution via a simple gradient algorithm. The idea is evaluated in supervised learning (regression) and generative modeling (VAE for images).

**Ethical Concerns:**

I don't have any remaining omitted ethical concerns regarding this paper.

**Limitations And Societal Impact:**

I believe the authors have adequately addressed societal impact & limitations of their work.

**Main Review:**

While the idea presented in this paper seems reasonable, the paper comes off as incomplete scientific work as it lacks comparisons against many related methods that have been developed in this area (all using pseudo-inputs for better uncertainty). Since the paper does not present strong theoretical or conceptual advantages of their proposed method, a reader must be convinced of its merits purely from the empirical comparisons, which are currently insufficient to convince me in this paper.

I find the related work too general and misleadingly presents the history as if nobody has already considered out-of-distribution pseudo-inputs training to improve uncertainty estimates.
When in fact there are tons of related works that have been overlooked such as:

Maximizing Overall Diversity for Improved Uncertainty Estimates in Deep Ensembles.
Jain et al. 2020.
https://arxiv.org/abs/1906.07380

This paper also generates augmented data to improve uncertainty estimates by increasing their entropy at the augmented samples, and does not require a GAN. It even explicitly considered generating pseudo-inputs with low density as in your work. Should be conceptually and empirically compared against as it seems highly related.

Noise Contrastive Priors for Functional Uncertainty.
Hafner et al. 2019
https://arxiv.org/abs/1807.09289

Also Lee et al. ("Confidence-calibrated...") should be contrasted against far more, currently the only clear difference I can understand from reading your paper is that you do not require a GAN while Lee et al do, but I'm sure there are other key differences that should be described. The authors write that Lee et al require an extra GAN as a downside, but do not acknowledge their method requires an extra density estimator. Furthermore there is no discussion in this paper of the finding by Lee et al that "boundary samples" correspond to the best pseudo-inputs, whereas how close to the "boundary" pseudo inputs from your method will lie is unclear and may arbitrarily depend on hyperparameter values.


Other related work includes:

Prediction-Based Regularization Using Data Augmented Regression.
Hooker et al. 2012.
http://faculty.bscb.cornell.edu/~hooker/darpaper.pdf

Predictive Uncertainty Estimation via Prior Networks.
Malinin et al. 2018
https://arxiv.org/abs/1802.10501

Uncertainty in Neural Networks: Approximately Bayesian Ensembling
Pierce et al. 2020.
https://arxiv.org/abs/1810.05546

Furthermore there are a suspicious number of nan reported results for baseline methods, for which it is unclear why this is the case (and again suggests the chosen baselines in this work are not properly intended for the tasks considered).

Also how did you choose hyperparameters for baselines and your model in each task? For example in the MNIST vs FashionMNIST experiment, it seems possible to tweak hyperparameters to obtain the favorable AUROC curve depicted in Fig 10. There should be many more pairs of image datasets evaluated in this experiment using the same hyperparameters to control for hyperparameter settings and more convincingly demonstrate the superiority of your proposed method.

Finally your method depends on a density estimator, so it would be good to show your chosen density estimator alone does not lead to good OOD detection and your improved VAE is truly needed. For example, perhaps a simple linear combination of density-estimate + standard VAE is just as good for OOD detection in which case your method would not really be necessary.

### Update after reading author response to my original review ###

I believe the paper will be much stronger after the inclusion of the additional baselines and major updates to the related work to properly reflect the historical context of this contribution. However I think this stronger paper should be submitted to the next conference for a proper review, as there are simply too many critical updates made in the revision for me to recommend acceptance based only on the brief descriptions of the changes through the comment system. In particular, the revision needs to be carefully written in such a way that precisely describes what previously existed before and what exactly is a new contribution in this paper, without over-claiming anything.


**Time Spent Reviewing:**

4

---

> ### Author Response · Authors · 2021-08-10
> **Reply (part I)**
>
> Thank you very much for your thorough review and the relevant references you mentioned, which we indeed had missed.
>
> **Since the paper does not present strong theoretical or conceptual advantages of their proposed method**
>
> We view the conceptual advantages of the proposed method as follows:
> 1. Analytical formulation of the model uncertainty, with a direct distinction between the aleatoric and epistemic uncertainty.
> 2. Improve consistently the robustness and reliability of variance predictions, with controlled extrapolation, leading to better sampling capabilities.
> 3. Generalisability and modularity. The training of the variance is decoupled from the training of the mean, and therefore does not require a degradation of the mean predictive power. Furthermore, the generation of pseudo-inputs, agnostic of the density estimator used, can be decoupled from the training of the model as well, and executed a priori, as long as we have access to the training inputs. (This is for example not the case in a VAE setting, but one can very well in that case first train the encoder and decoder mean with fixed decoder variance as is most commonly done, then generate pseudo-inputs using the distribution of latent encodings, and finally train the variational variance, reusing the pretrained encoder and decoder mean.)
>
> **I find the related work too general and misleadingly presents the history as if nobody has already considered out-of-distribution pseudo-inputs training to improve uncertainty estimates**
>
> Thanks to the relevant references you and other reviewers provided, we do recognise that our related work is incomplete and will extend our discussion of previous approaches for generating pseudo-inputs for out-of-distribution regularisation (see below).
>
> **Maximizing Overall Diversity for Improved Uncertainty Estimates in Deep Ensembles. Jain et al. 2020.**
>
> We agree that the overlooked methods in Maximizing Overall Diversity for Improved Uncertainty Estimates in Deep Ensembles (MOD), have some similarities to our work, and we acknowledge the relevance of the comparison. These methods generate pseudo-inputs by sampling uniformly over the input space, which we expect to scale poorly due to the curse of dimensionality. In contrast, our approach leverages the observed data and optimizes for pseudo-input placement.
>
> Unfortunately, a direct empirical comparison is non-trivial. The MOD code relies on a sigmoid-transformation [1] of the regression targets, which changes RMSE and log-likelihoods significantly. This effectively skews the results, which can also be seen in Table 6 of their paper as the RSME’s for the well known UCI datasets are reported very low. Surprisingly, a sigmoid-transformation [2] is also an integral part of the MOD model itself and cannot easily be dropped (doing so alters the MOD behavior significantly and it is unclear if the resulting method would be representative of the work).
>
> [1] https://github.com/tmfs10/bb-opt/blob/master/src/utils.py#L474-L476
>
> [2] https://github.com/tmfs10/bb-opt/blob/master/src/deep_ensemble_sid.py#L125
>
> **Noise Contrastive Priors for Functional Uncertainty. Hafner et al. 2019**
>
> Particularly close to our work are the Noise Contrastive Priors (NCPs) [Hafner et al., 2019] who also regularise their posterior distribution through a high-entropy prior on automatically generated OOD pseudo-inputs.
>
> Specifically, their method for generating pseudo-inputs, $\hat{x} = x + \epsilon$, with \epsilon is Gaussian noise, results in modelling the OOD distribution as the convolved Gaussian mixture distribution $p(\hat{x}) = 1/N \sum_{i}N(\hat{x}-x_i | 0, \sigma^2)$.
> Unaware of the work of Hafner et al. we started our exploration with exactly this idea, but observed empirically that it did not produce in practice pseudo-inputs that were actually out-of-distribution (this blog post http://krasserm.github.io/2020/09/25/reliable-uncertainty-estimates/ provides supporting arguments, as it clearly shows a significant overlap on a toy example between the training and OOD distributions), which hindered the performance of our model, due to a difficult tradeoff between over regularisation and unpredictable extrapolation (if sigma gets larger).
>
> To counteract this, we introduced Algorithm 1. which, through gradient descent of the density of training inputs pushes away the pseudo-inputs originally sampled as is done by Hafner et al. towards regions of low data density. In practice, because we cannot realistically evaluate and run automatic differentiation on the true complete convolved Gaussian mixture on large datasets, we instead approximate the OOD distribution with a Bayesian Gaussian Mixture with a limited number of components.
> Overall, we therefore view our proposal for generating pseudo-inputs as a generalisation of what is proposed by Hafner et al., which allows us to limit over-regularisation due to pseudo-inputs and increase the performance of the model, especially on high dimensional datasets.
>
> For reference, the following table compares the mean ELBO over 5 trials of our proposal with selected parameters for the gradient descent (dVV), with our proposal without any gradient descent (dVV - 0 steps) and our method with pseudo-inputs generated as Gaussian noise (dVV - Gaussian noise, as is done in Hafner et al.) on the UCI benchmark. We thus observe that our method performs better on a larger selection of experiments (boston, carbon, ccpp, concrete, energy, kin8nm, naval, superconduct - a total of 8/12 experiments).
>
> | Method | uci_boston | uci_carbon | uci_ccpp | uci_concrete | uci_energy | uci_kin8nm | uci_naval | uci_protein | uci_superconduct | uci_wine_red | uci_wine_white | uci_yacht |
> |------------|----------------|----------------|-------------|-------------------|-----------------|-----------------|--------------|----------------|--------------------------|-------------------|----------------------|---------------|
> | dVV      | -0.6144515 | 1.19136671 | -0.1410655 | -0.4431365 | 0.66507459 | -0.2474278 | 0.52161705 | -1.3193981 | -0.5427999 | -1.9969132 | -1.8216372 | -17.59926 |
> | dVV - 0 steps | -0.6584299 | 1.22908062 | -0.1306134 | -0.4325904 | 0.65596978 | -0.2549232 | 0.49647621 | -1.3166391 | -0.5578889 | -2.1916367 | -2.0302405 | -21.801952 |
> | dVV - Gaussian noise | -0.8194847 | -0.4729474 | -0.5693008 | -0.645149 | -0.5147148 | -0.5907792 | -0.5367214 | -1.2355864 | -0.7236105 | -1.7473126 | -1.5542871 | -0.5071667 |
>
> Furthermore, the method developed by Hafner et al. focuses on the regularisation of Bayesian Neural Networks. They specifically only impose a distribution on the last layer of a bifurcated mean variance network. With this regard they decompose the model uncertainty between the aleatoric component, as the output of the variance network, and the epistemic component as the uncertainty of the distribution of the predicted mean. Because of this design choice, the OOD regularisation rests on the minimisation of the KL divergence between the distribution of the predicted means and a prior over the entire data space (that is {input,output} domain), which forces the assumption that generated pseudo-inputs share the same outputs as the inputs from which they were generated. In our method, thanks to our censoring approach, we can avoid the delicate problem that arises when one needs to model the pseudo-outputs associated with pseudo-inputs. Indeed, who can tell what is the correct output for inputs that don’t have an observed output ?
>
> **Also Lee et al. ("Confidence-calibrated...") should be contrasted against far more**
>
> We acknowledge the influence of the work of Lee et al. had in our research, and that we might have highlighted the downside resulting from the reliance of their method on a GAN with more ease than ours on a density estimate. Nevertheless, this reasoning is supported by the greater overhead and practical complexity incurred by training a GAN for a simple regression or classification task compared to a simple density estimate (e.g (Bayesian) Gaussian Mixture). To be transparent, we felt like having to train a much more complex model (GAN) to generate pseudo-inputs than what is required to solve the original task at hand is not satisfactory.
>
> We further believed that Figure 2., and the introductory paragraph of section 2.2 would be faithful to and echo the intuition of Lee et al. regarding the placement of pseudo-inputs at the boundary of the training data, and acknowledge that we could make this clearer by specifying that “low-density regions of the input space that neighbours training data” extend the notion of boundary to more complex input distributions.
>
> Supporting evidence regarding the benefits of placing pseudo-inputs “close” to the training distribution can be found in the following table that compares the mean measured OOD KL divergence over 5 trials of three methods, No OOD regularisation (original variational variance - VV), “close” pseudo-inputs OOD regularisation (ours - dVV) and “far-away” pseudo-inputs regularisation (generated with Gaussian noise as in Hafner et al. but with a large noise variance \sigma = 15). Clearly, only the pseudo-inputs generated at the boundary of the input distribution allow for a reliable (in the sense of matching the prior) extrapolation.
>
> | method | uci_ccpp | uci_wine_white |
> |-----------|--------------|----------------------|
> | No ood (vv) | 0.65483451 | 1.71848818 |
> | Close ood (dVV) | 0.01864026 | 0.07170208|
> | Far ood (gaussian noise, large noise) | 0.02282202 | 0.94072956 |

---

> > ### Author Response · Authors · 2021-08-10
> > **Reply (part II)**
> >
> > **there are a suspicious number of nan reported results for baseline methods**
> >
> > First, the reported nans, mostly on the evaluated ELBO or RMSE between generated samples and true targets, emerge from the impossibility of evaluation of these metrics (e.g ELBO is not defined for non Bayesian models) or from the absence of reported metrics from legacy code found on open repositories.
> >
> > We specifically re-used public code from Skafte et al.. Reliable training and estimation of variance networks.NeurIPS 2019. which itself copied code from the other baselines, which did not allow for direct computation of the missing metrics.
> >
> > **Also how did you choose hyperparameters for baselines and your model in each task?**
> >
> > The hyperparameters chosen for the baselines were the ones given in the open repository of Skafte et al. (see above).
> >
> > For our model, the hyperparameters can be decomposed into two classes:
> >
> > 1. The PIG hyperparameters were chosen through a coarse grid search on a sub selection of experiments from the UCI benchmarks, and applied to the rest of the experiments. In practice we observed that Algorithm 1 is relatively insensitive to hyperparameters.
> > For example the first table below describes the evolution of the ELBO and the log-likelihood as a function of the number of steps on the carbon dataset.
> > The ELBO and LLK here have opposite evolutions based on the number of steps - hence our practical choice
> > of a “middle ground” with n = 5.
> > And the second table depending on the threshold reveals a similar trend as for the number of steps.
> > The results are naturally varying depending on the dataset chosen, as different overall input density will more or less allow the gradient descent iterations to modify the original distribution of noisy pseudo-inputs. We regard this as a strength of our proposed method, if the distribution of pseudo-inputs can be improved and better separated from the training input distribution, it will leverage this possibility, but if it’s not the case, pseudo inputs will not vary from their original position, generated as if they were inputs perturbed by Gaussian noise.
> >
> > | N steps | ELBO | LLK |
> > |------------|---------|--------|
> > | 0            | 1.25268706 | 1.35993608 |
> > | 1            | 1.23172506 | 1.36138062 |
> > | 5            | 1.19150706 | 1.42261565 |
> > | 10          | 1.18344955 | 1.43695784 |
> >
> >
> > | Threshold | ELBO | LLK |
> > |---------------|-----------------|----------------|
> > | 0.1            | 1.19291011 | 1.4178112 |
> > | 0.01          | 1.19224445 | 1.41748428 |
> > | 0.001        | 1.25230094 | 1.35911342 |
> >
> >
> > 2. The prior parameters were chosen according to the procedure described in II.6 & III.5 of the supplementary materials. For regression, an empirical study on both the toy regression model and on the UCI benchmarks revealed to us model instabilities for alpha < 1.5, as is also reported by Stirn & Knowles. Variational variance: Simple and reliable predictive variance parameterization. arXiv:2006.04910, 2020. and led us to adopt the value alpha=1.5 and a value for beta that matches the empirical variance of each dataset. For the generative case, we adopted either a similar approach in the general case, while in the case of out-of-distribution detection we chose an heteroscedastic prior whose uncertainty would increase further away from data to reinforce the regularisation effect.
> >
> > Thank you for your suggestion to include other pairs of in-distribution / out-of-distribution to demonstrate that the chosen hyperparameters work for a variety of cases. We will include such datasets similar to MNIST (omniglot, NotMNIST) in our annex. We nevertheless know that our method should also significantly improve the VAE’s ability to predict OOD samples with a lower likelihood, as, as we have seen just above, its performance is relatively insensitive to hyperparameters, and as prior parameters are automatically derived from the training dataset.
> >
> > **For example, perhaps a simple linear combination of density-estimate + standard VAE is just as good for OOD detection in which case your method would not really be necessary**
> >
> > This is a valid concern, but we did observe in practice, and as was reported in several other works as well (see e.g. Havtorn et al. Hierarchical vaes know what they don’t know, ICML 2021. ) that without our method, the encodings of out-of-distribution inputs are intermixed with the encodings of in-distribution inputs. This would without doubt lead to high density estimates for out-of-distribution encodings that would as a result be hard to distinguish from the in-distribution encodings.
> > Thanks to the OOD detection experiment we aimed to demonstrate that a well-behaved decoder variance allows the encoder to then be re-trained to behave sensibly (in the sense that it would encode differently in-distribution and out-of-distribution data) on OOD data. We are, thus, hesitant to include the suggested experiment as it could be perceived as an artificially easy baseline.

---

> ### Author Response · Authors · 2021-08-23
> **Follow up**
>
> We appreciate the review update, and sympathize with the concern that the updates are sufficiently many that a new round of reviews are in order. We do want to raise the following counter-arguments:
>
> * The key changes to the paper are in the Related Works section and are, thus, fairly isolated.
> * Our work originated with the same idea as the *noise contrastive priors*, but we empirically found this to over-regularize the mean, which motivated the presented algorithm. That this initial idea was previously explored by Hafner et al. only serves to strengthen the motivation of our work. In our view, the missed related work thus only improves our paper, and it is in our own best interest to leverage this in a revision.
> * If the paper is resubmitted to a different venue, it will get a new set of reviewers who will push the paper in directions of their own. So if you like the paper with our suggested changes, we ask you to please reconsider.

---

### Decision · Program_Chairs · 2021-09-27

**Decision:**

Reject

**Comment:**

The idea of the paper is interesting, however the submitted work fails to acknowledge a fair amount of related work, and the experimental evaluation fails to incorporate sufficient relevant baselines (of which there are many). Although the authors attempted to address this in rebuttal, incorporating this into the paper would require significant changes. The reviewers also noted a number of other concerns (more extensive experiments, clarity) that can be addressed at the same time. Thus I recommend the authors revise their manuscript and resubmit with these improvements.